# SpargeAttention: Accurate and Training-free Sparse Attention Accelerating Any Model Inference

**Jintao Zhang**[* 1] **Chendong Xiang**[* 1] **Haofeng Huang**[* 1 2] **Jia Wei**[1] **Haocheng Xi**[3] **Jun Zhu**[1] **Jianfei Chen**[1]

https://github.com/thu-ml/SpargeAttn

## Abstract

An efficient attention implementation is essential for large models due to its quadratic time complexity. Fortunately, attention commonly exhibits sparsity, i.e., many values in the attention map are near zero, allowing for the omission of corresponding computations. Many studies have utilized the sparse pattern to accelerate attention. However, most existing works focus on optimizing attention within specific models by exploiting certain sparse patterns of the attention map. **A universal sparse attention that guarantees both the speedup and end-to-end performance of diverse models remains elusive.** In this paper, we propose SpargeAttn, a universal sparse and quantized attention for any model. Our method uses a two-stage online filter: in the first stage, we rapidly and accurately predict the attention map, enabling the skip of some matrix multiplications in attention. In the second stage, we design an online softmax-aware filter that incurs no extra overhead and further skips some matrix multiplications. Experiments show that our method significantly accelerates diverse models, including language, image, and video generation, without sacrificing end-to-end metrics. The code is available at https://github.com/thu-ml/SpargeAttn. [1]

*Equal contribution [1]Dept. of Comp. Sci. and Tech., Institute for AI, BNRist Center, THBI Lab, Tsinghua-Bosch Joint ML Center, Tsinghua University [2]Institute for Interdisciplinary Information Sciences, Tsinghua University [3]EECS, University of California, Berkeley. Correspondence to: Jun Zhu <dc-szj@mail.tsinghua.edu.cn>.

*Proceedings of the 42ⁿᵈ International Conference on Machine Learning*, Vancouver, Canada. PMLR 267, 2025. Copyright 2025 by the author(s).

[1]All experiments using SpargeAttn is based on SageAttention. An updated implementation based on SageAttention2, is available at https://github.com/thu-ml/SpargeAttn. It further offers a 30% speedup over the attention in this paper.

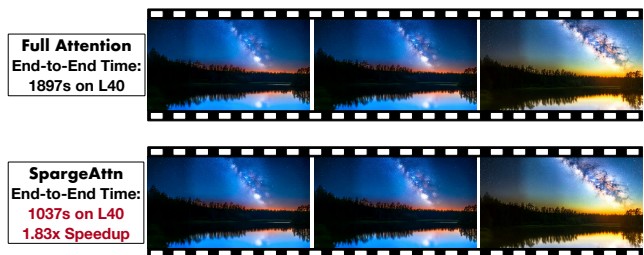

*Figure 1.* SpargeAttn can achieve 1.83x speedup on Mochi on L40 GPU, with no video quality loss.

## 1. Introduction

As sequence lengths in large models become longer, such as 45K-128K in video generation and language models (Yang et al., 2025b; Bao et al., 2024; Dubey et al., 2024; Zhang et al., 2025c), the time consuming of attention occupies a significant portion of inference latency in large models (Zhang et al., 2025d). Fortunately, the attention map $P = \text{Softmax}(QK^\top/\sqrt{d})$ exhibits inherent sparsity, as the softmax operation often creates many values approaching zero (Deng et al., 2024). *Sparse attention* exploit such sparsity to accelerate attention by (1) constructing a "sparse mask", which indicates the important non-zero entries of the attention map $P$ that should be computed, and (2) computing attention only for the parts corresponding to the *sparse mask*. There are three distinct categories of sparse attention methods based on how the sparse mask is generated. *pattern-based method* (Zhang et al., 2023; Xiao et al., 2024a; Fu et al., 2024; Zhu et al., 2024; Xiao et al., 2025; 2024b; Zhang et al., 2025h) relies on specific sparsity patterns based on empirical observations, *dynamic sparse attention* (Ribar et al., 2024; Singhania et al., 2024; Jiang et al., 2024; Lai et al., 2025; Gao et al., 2024; Xi et al., 2025; Yang et al., 2025a; Zhang et al., 2025f) computes the mask on-the-fly based on the inputs, and *training-based method* (Kitaev et al., 2020; Pagliardini et al., 2023; Zhang et al., 2025i) directly train models with native sparse attention.

**Limitation**. (L1. Universality) Though existing sparse attention methods already demonstrate promising speedup on some tasks, their universality is still limited. Existing works

are typically developed for specific tasks, such as language modeling, utilizing task-specific patterns such as sliding windows or attention sinks. However, the attention pattern varies significantly across tasks (see examples in Fig. 2), making these patterns hard to generalize. (L2. Usability) Moreover, it is difficult to implement both *accurate* and *efficient* sparse attention for any input. This is because *accuracy* demands precise prediction of the sparse regions in the attention map, while *efficiency* requires the overhead of this prediction to be minimal. However, current methods are difficult to effectively satisfy both of the requirements simultaneously. For example, MInference (Jiang et al., 2024) requires a large sequence length, such as 100K, to achieve a noticeable speedup.

**Goal.** We aim to design a training-free sparse attention operator that accelerates all models without metrics loss.

**Our approach.** In this work, we develop `SpargeAttn`, a *training-free* sparse attention that can be adopted *universally* on various tasks, including language modeling and text-to-image/video, and various sequence lengths. We propose three main techniques to improve the universality, accuracy, and efficiency. First, we propose a universal sparse mask prediction algorithm, which constructs the sparse mask by compressing each block of $Q$, $K$ to a single token. Importantly, we compress *selectively* based on the *similarity* of tokens within the block, so the algorithm can accurately predict sparse masks universally across tasks. Second, we propose a sparse online softmax algorithm at the GPU warp level, which further omits some $PV$ products by leveraging the difference between global maximum values and local maximum values in online softmax. Third, we integrate this sparse approach into the 8-bit quantized SageAttention framework for further acceleration. To the best of our knowledge, `SpargeAttn` is the **first** sparse attention method that can actually accelerate across language, image, and video models without compromising accuracy.

**Result.** We evaluate `SpargeAttn` on a variety of generative tasks, including language modeling and text-to-image/video, with comprehensive performance metrics on the model quality. `SpargeAttn` can robustly retain model end-to-end performance while existing sparse attention baselines incur degradation. Moreover, `SpargeAttn` is 2.5x to 5x faster than existing dense and sparse attention models.

## 2. Related Work

Depending on how the sparsity mask is constructed, sparse attention methods can be divided into three types: **(1) Pattern required methods** rely on some fixed patterns of the attention map, such as sliding windows or attention sinks (Xiao et al., 2024b). H2O (Zhang et al., 2023), InfLLM (Xiao et al., 2024a), and DUOAttention (Xiao

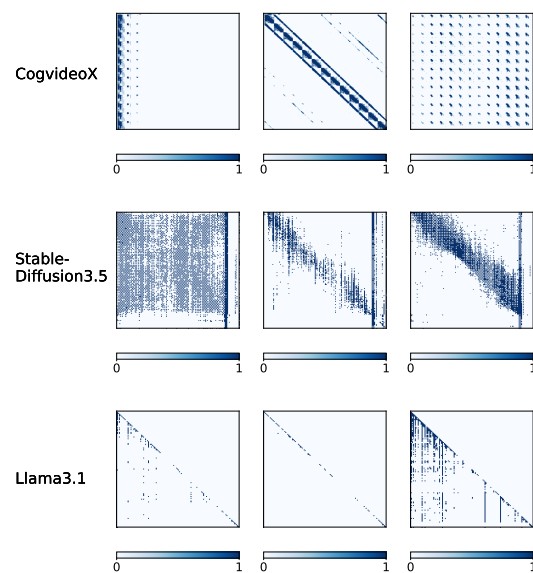

*Figure 2.* Some sampled patterns of attention map $P$ in video, image, and language generation models.

et al., 2025) rely on sliding window pattern. SampleAttention (Zhu et al., 2024), MOA (Fu et al., 2024), and StreamingLLM (Xiao et al., 2024b) rely on sliding window and attention sink pattern. DitFastAttn (Yuan et al., 2024) relies on sliding window patterns and similarities between different attention maps. Moreover, DitFastAttn is restricted to simple diffusion transformers, showing incompatibility with language models and MMDiT models like Flux (Black Forest Labs, 2023), Stable Diffusion3 and 3.5 (Stability AI, 2023), and CogVideoX (Yang et al., 2025b). As the pattern varies across models, these methods may not universally work for different models. **(2) Dynamic sparse methods** dynamically construct the sparse mask based on the input without the need of preset patterns, and are thus potentially more universal. Existing works can be further categorized into channel compression and token compression. Channel compression methods include SparQAttn (Ribar et al., 2024) and LokiAttn (Singhania et al., 2024). They construct the mask by carrying full attention with reduced dimensionality. However, as the dimension is already small, e.g., 64, 128, in commonly used attention, the speedup potential might be limited. Token compression methods include MInference (Jiang et al., 2024) and FlexPrefill (Lai et al., 2025). They construct the mask by compressing each block of tokens to a single token and compute attention on this shorter sequence. However, this approximation is too aggressive: missing important blocks of $P$ is possible if they do not have a large attention score on the compressed sequence. SeerAttention (Gao et al., 2024) requires training of additional parameters for attention, which is expensive to use. Moreover, they are all designed for language models, and their applicability to other model types, such as diffusion models, remains uncertain. **(3) Training-based methods**

modify the attention computation logic, requiring retraining the entire model, such as Reformer (Kitaev et al., 2020) and FastAttention (Pagliardini et al., 2023). These methods are much more expensive to use than training-free methods.

There are other ways to accelerate attention, such as optimizing the kernel implementation (Dao et al., 2022; Dao, 2024; Shah et al., 2024), quantization (Zhang et al., 2025d), distributing the workload (Liu et al., 2024a), and designing linear time attention (Wang et al., 2020; Choromanski et al., 2021; Yu et al., 2022; Katharopoulos et al., 2020). They are orthogonal to our approach.

## 3. `SpargeAttn`

`SpargeAttn` contains a two-stage online filter to implement sparse FlashAttention. First, as shown in `Step1` and `Step2` in Fig. 3, we design a fast and accurate method to predict the sparse block in the attention map, thereby skipping the corresponding products of $Q_i K_j^\top$ and $\widetilde{P}_{ij} V_j$. Second, as shown in `Step3` in Fig. 3, we design a sparse online softmax method to further skip the products of $\widetilde{P}_{ij} V_j$.

### 3.1. Sparse FlashAttention

`SpargeAttn` adopts the tiling strategy of FlashAttention (Dao, 2024), and skip computing the blocks that are filtered out. Consider an attention operation $S = QK^\top/\sqrt{d}$, $P = \sigma(S)$, $O = PV$, where $\sigma(S)_{ij} = \exp(S_{ij})/\sum_k \exp(S_{ik})$ is the softmax operation. Let $N$ be the sequence length and $d$ be the dimensionality of each head; the matrices $Q$, $K$, and $V$ each have dimensions $N \times d$, while the matrix $S$ and $P$ is $N \times N$. FlashAttention proposes to tile $Q$, $K$, and $V$ from the token dimension into blocks $\{Q_i\}, \{K_i\}, \{V_i\}$ with block sizes $b_q$, $b_k$, $b_k$, respectively. Then, it uses online softmax (Milakov & Gimelshein, 2018) to progressively compute each block of $O$, i.e., $O_i$:

$$S_{ij} = Q_i K_j^\top/\sqrt{d}, \ \ (m_{ij}, \widetilde{P}_{ij}) = \tilde{\sigma}(m_{i,j-1}, S_{ij}),$$
$$l_{ij} = \exp(m_{i,j-1} - m_{ij})l_{i,j-1} + \text{rowsum}(\widetilde{P}_{ij}),$$
$$O_{ij} = \text{diag}\left(\exp(m_{i,j-1} - m_{ij})\right)O_{i,j-1} + \widetilde{P}_{ij} V_j \quad (1)$$

where $m_{ij}$ and $l_{ij}$ are $b_q \times 1$ vectors, which are initialized to $-\infty$ and 0 respectively. The $\tilde{\sigma}()$ is an operator similar to softmax.: $m_{ij} = \max\{m_{i,j-1}, \text{rowmax}(S_{ij})\}$, $\widetilde{P}_{ij} = \exp(S_{ij} - m_{ij})$. Finally, the output $O_i$ can be computed by $O_i = \text{diag}(l_{ij})^{-1} O_{ij}$.

Implementing sparse FlashAttention is intuitive. By *skipping* certain block matrix multiplications of $Q_i K_j^\top$ and $\widetilde{P}_{ij} V_j$, we can accelerate the attention computation. We formulate sparse attention based on FlashAttention in the following definitions.

**Definition 1** (Block Masks). Let $M_g$ and $M_{pv}$ be binary masks of dimensions $\lceil N/b_q \rceil \times \lceil N/b_k \rceil$, where each value

is either 0 or 1. These masks determine which computations are skipped in the sparse attention mechanism.

**Definition 2** (Sparse FlashAttention). The computation rules for sparse FlashAttention based on the masks are defined as follows:

$$Q_i K_j^\top, \widetilde{P}_{ij} V_j \ \text{ are skipped if } \ M_g[i, j] = 0. \quad (2)$$

$$\widetilde{P}_{ij} V_j \ \text{ is skipped if } \ M_{pv}[i, j] = 0. \quad (3)$$

### 3.2. Selective Token Compression for Sparse Prediction

Key idea. Although attention maps vary across models, we observe that various models exhibit a common trait: Most neighboring tokens in the query and key matrices of the attention show high similarity (See Fig. 4). Consequently, for blocks composed of highly similar tokens, we can consolidate these tokens into a single representative token for the block. Based on this observation, we propose a pattern-free online prediction method for identifying sparse blocks in $P$ to skip some computation of $Q_i K_j^\top$ and $\widetilde{P}_{ij} V_j$ during the FlashAttention process. Specifically, we first compress blocks exhibiting high self-similarity within $Q$ and $K$ into tokens. Then, we swiftly compute a compressed attention map $\hat{P}$ using the compressed $Q$ and $K$. Finally, we selectively compute $\{Q_i K_j^\top, \widetilde{P}_{ij} V_j\}$ for those pairs $(i, j)$ where $\{\hat{P}[i, j]\}$ accumulates a high score in the compressed attention map. Notably, block selection was only performed in the high self-similarity blocks, which we also refer to as "selective blocks." For those non-self-similar blocks, as a good presentation token for the whole block is hard to find, we choose to always compute the non-self-similar block in the attention operation, which we also refer to as "fix blocks." Importantly, compressing only the token blocks with high self-similarity is crucial, as omitting computations for fix blocks can result in the loss of critical information. This will be confirmed in Sec. 4 and A.2.

Prediction. As shown in `Step1` in Fig. 3, we first compute a mean cosine similarity across tokens for each block of $Q$ and $K$. Next, we compress each block into a single token by calculating a mean across tokens. Then, we compute a compressed $QK^\top$ using the compressed $Q$ and $K$. Finally, to prevent interference from non-self-similar blocks, i.e., the block similarity less than a hyper-parameter $\theta$, we set the corresponding values in $S$ to $-\infty$, and then obtain a compressed attention map through softmax. This algorithm can be expressed as:

$$q = \{q_i\} = \{\text{mean}(Q_i, \text{axis} = 0)\}$$
$$k = \{k_j\} = \{\text{mean}(K_j, \text{axis} = 0)\}$$
$$s_{qi} = \text{CosSim}(Q_i), s_{kj} = \text{CosSim}(K_j)$$
$$\hat{S}[i] = q_i k^\top; \quad \hat{S}[:, j] = -\infty, \text{ If } s_{kj} < \theta$$
$$\hat{P}[i] = \text{Softmax}(\hat{S}[i])$$

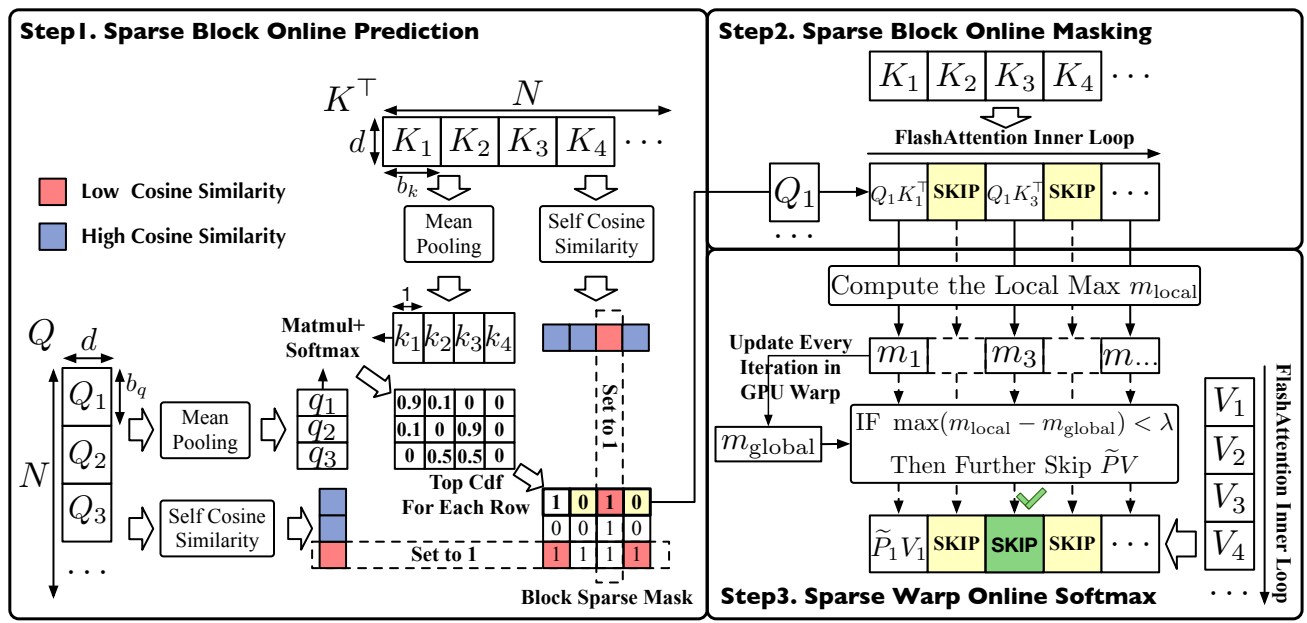

*Figure 3.* Workflow of `SpargeAttn`.

---

**Algorithm 1** Implementation of `SpargeAttn`.

---

1: **Input:** Matrices $Q(\text{FP16}), K(\text{FP16}), V(\text{FP16}) \in \mathbb{R}^{N \times d}$, block size $b_q, b_{kv}$, count of GPU Warps $c_w$, hyper-parameters $\tau, \theta, \text{and } \lambda$.
2: Divide $Q$ to $T_m = N/b_q$ blocks $\{Q_i\}$; divide $K, V$ to $T_n = N/b_{kv}$ blocks $\{K_i\}$ and $\{V_i\}$.
3: $\hat{Q}_i, \hat{K}_j, \delta_Q, \delta_K = \text{Quant}(Q_i, K_j)$ ; // per-block quantization in SageAttention.
4: $q = \{q_i\} = \{\text{mean}(Q_i, \text{axis} = 0)\}$ ; $\quad k = \{k_j\} = \{\text{mean}(K_j, \text{axis} = 0)\}$ ;
5: $\hat{S} = qk^\top$; $\quad s_{qi} = \text{CosSim}(Q_i)$; $\quad s_{kj} = \text{CosSim}(K_j)$; $\quad \hat{S}[:, j] = -\infty, \text{If } s_{kj} < \theta$;
6: $\hat{P}[i] = \text{Softmax}(\hat{S}[i])$ ; $\quad M[i, :] = \text{TopCdf}(\hat{P}[i], \tau)$ ; $\quad M[i, :] = 1, \text{If } s_{qi} < \theta$ ; $\quad M[:, j] = 1, \text{If } s_{kj} < \theta$ ;
7: **for** $i = 1$ **to** $T_m$ **do**
8: $\quad$ Load $\hat{Q}_i$ and $\delta_Q[i]$ into a SM ;
9: $\quad$ **for j** in $[1, T_n]$ **do**
10: $\quad\quad$ **if** $M[i, j]! = 0$ **then**
11: $\quad\quad\quad$ Load $\hat{K}_j, \hat{V}_j$, and $\delta_K[j]$ into the SM ;
12: $\quad\quad\quad$ $S_{ij} = \text{Matmul}(\hat{Q}_i, \hat{K}_j^T) \times \delta_Q \times \delta_K$ ; // dequantization of SageAttention.
13: $\quad\quad\quad$ $m_{\text{local}} = \text{rowmax}(S_{ij})$; $\quad m_{ij} = \max(m_{i,j-1}, m_{\text{local}})$; $\quad \widetilde{P}_{ij} = \exp(S_{ij} - m_{ij})$; $\quad l_{ij} = e^{m_{i,j-1} - m_{ij}} + \text{rowsum}(\widetilde{P}_{ij})$ ;
14: $\quad\quad\quad$ $i_w = \text{range}(c_w)$ ; $\quad I_w = [\frac{i_w * b_q}{c_w} : \frac{(i_w+1) * b_q}{c_w}]$ ;
15: $\quad\quad\quad$ **if** $\max(m_{\text{local}}[I_w] - m_{ij}[I_w]) > \lambda$ **then**
16: $\quad\quad\quad\quad$ $O_{ij}[I_w] = \text{diag}(e^{m_{i,j-1}[I_w] - m_{ij}[I_w]})^{-1} O_{i,j-1}[I_w] + \text{Matmul}(\widetilde{P}_{ij}[I_w], V_j)$ ; // Paralleled by $c_w$ warps.
17: $\quad\quad\quad$ **end if**
18: $\quad\quad$ **end if**
19: $\quad$ **end for**
20: $\quad$ $O_i = \text{diag}(l_{i,T_n}^\top)^{-1} O_{i,T_n}$ ;
21: $\quad$ Write $O_i$ ;
22: **end for**
23: **return** $O = \{O_i\}$ ;

---

where $Q_i \in \mathbb{R}^{b_q \times d}, q_i \in \mathbb{R}^{1 \times d}, K_j \in \mathbb{R}^{b_k \times d}, k_j \in \mathbb{R}^{1 \times d}$ and $\text{CosSim}(X) = mean(\frac{XX^\top}{|\max(XX^\top)|})$ measures the cosine-similarity within a block.

For each row of $\hat{P}$, i.e., $\hat{P}[i]$, we select the positions of the top values whose cumulative sum reaches $\tau \cdot \sum \hat{P}[i]$, where $\tau$ is a hyper-parameter. These positions are set to 1

in $M_g[i, :]$, while all other positions are set to 0.

$$M_g[i, :] = \text{TopCdf}(\hat{P}[i], \tau) \quad (4)$$

where the $\text{TopCdf}(\hat{P}[i], \tau)$ can be formulated as follows.

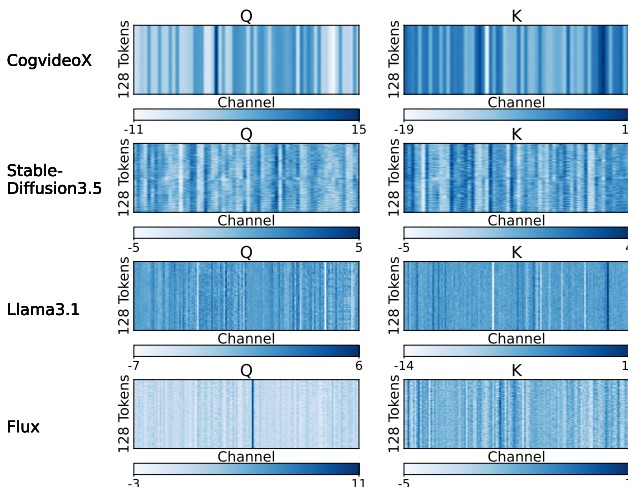

*Figure 4.* Exemplary patterns of the query and key in the attention of various models.

```
def Top_Cdf(P[i], tau):
    sorted_P, idx = torch.sort(P[i], descending=True)
    cusum_P = torch.cumsum(sorted_P, dim=0)
    mask = cusum_P <= tau * P[i].sum()
    M_i = torch.zeros_like(mask)
    M_i[idx] = mask
    return M_i
```

Finally, we need to ensure that calculations involving non-self-similar blocks(fix block) of $Q$ or $K$ are not omitted. Therefore, we set all values in the rows of $M_g$ corresponding to not self-similar blocks of $Q$ to 1, and all values in the columns of $M_g$ corresponding to non-self-similar blocks of $K$ to 1.

$$M_g[i,:] = 1, \text{ If } s_{qi} < \theta; \quad M_g[:,j] = 1, \text{ If } s_{kj} < \theta \quad (5)$$

### 3.3. Masking of the First Stage

Masking. The $M_g$ can be applied in FlashAttention directly to save some computation. In the inner loop of FlashAttention, i.e., during computing attention between a $Q_i$ and $\{K_j\}, \{V_j\}$, we can skip $\{Q_i K_j^\top, \widetilde{P}_{ij} V_j\}$ when $M_g[i,j] = 0$.

$$\text{Skip } Q_i K_j^\top \text{ and } \widetilde{P}_{ij} V_j, \text{ If } M_g[i,j] = 0 \quad (6)$$

### 3.4. Sparse Warp Online Softmax

Key idea. We can further identify the small enough values in the attention map during the online softmax process. If all values in $\widetilde{P}_{ij}$ are close enough to zero, the $\widetilde{P}_{ij} V_j$ will be negligible and can be omitted.

To identify which $\widetilde{P}_{ij} = \exp(S_{ij} - m_{i,j})$ (See Sec. 3.1) contains values small enough to be omitted, we note that in every inner loop of FlashAttention, the $O_{ij}$ will be scaled

by $\exp(m_{i,j-1} - m_{ij})$ and then plus the $\widetilde{P}_{ij} V_j$:

$$m_{\text{local}} = \text{rowmax}(S_{ij}), \quad m_{ij} = \max\{m_{i,j-1}, m_{\text{local}}\}$$
$$O_{ij} = \text{diag}\left(\exp(m_{i,j-1} - m_{ij})\right) O_{i,j-1} + \widetilde{P}_{ij} V_j$$

If $\text{rowmax}(S_{ij}) < m_{ij}$, then $m_{ij} = m_{i,j-1}$. Consequently, $O_{ij} = O_{i,j-1} + \widetilde{P}_{ij} V_j$. Furthermore, if $\text{rowmax}(S_{ij}) \ll m_{ij}$ holds true, then all values in $\widetilde{P}_{ij} = \exp(S_{ij} - m_{ij})$ are close to 0. This results in all values in $\widetilde{P}_{ij} V_j$ being close to 0. This condition implies that $\widetilde{P}_{ij} V_j$ is negligible when $\text{rowmax}(S_{ij})$ is significantly smaller than $m_{ij}$:

$$O_{ij} \approx O_{i,j-1}, \quad \text{if } \max\left(\exp(S_{ij} - m_{ij})\right) \to 0$$
$$\max\left(\exp(S_{ij} - m_{ij})\right) \to 0 \Leftrightarrow \max(m_{\text{local}} - m_{ij}) < \lambda$$

The above equivalence is satisfied when $\lambda$ is small enough.

Therefore, based on the analysis above, we propose a simple yet effective sparse method to further skip the $\widetilde{P}_{ij} V_j$ computation. Specifically, in the inner loop of FlashAttention, the $S_{ij}$ will be split by $c_w$ GPU warps to $\{S_{ij}[\frac{i_w * b_q}{c_w} : \frac{(i_w+1) * b_q}{c_w}, :]\}$, where $i_w$ is the index of the GPU warp. Let $I_w = [\frac{i_w * b_q}{c_w} : \frac{(i_w+1) * b_q}{c_w}]$. If $\max(m_{\text{local}}[I_w] - m_{ij}[I_w]) < \lambda$, where $\lambda$ is small enough, then $O_{ij}[I_w] \approx O_{i,j-1}[I_w]$, and we will skip the computation of $\widetilde{P}_{ij}[I_w] V_j$ which is used to update $O_{ij}[I_w]$.

### 3.5. Combined with SageAttention

To further accelerate our implementation of sparse attention, we integrate our method into SageAttention (Zhang et al., 2025a;d;g;b;e), which proposes a quantized method for accelerating attention. Since quantization (Hu et al., 2025; Zhang et al., 2025j) operations and sparse operations are orthogonal, sparse computation can be directly applied to SageAttention. The complete algorithm is shown in Algorithm 1. Specifically, first, we need to add one judgment at the beginning of the inner loop of SageAttention (Line 10, Algorithm 1) to decide whether to skip the whole inner loop once. Second, we add another judgment before the updating of $O_{ij}$ in the inner loop of SageAttention (Line, in Algorithm 1) to decide whether to skip the computation of $\widetilde{P}_{ij} V_j$. Moreover, to minimize the attention map prediction overhead, we implement the prediction using CUDA and adopt some kernel fusion techniques.

### 3.6. Hyper-parameters Determination for Model Layer

Based on the method description in Sec. 3.2 and 3.4, our method incorporates three hyper-parameters: $\tau \in (0, 1)$, $\theta \in (-1, 1)$, and $\lambda < 0$. The parameter determination process for each attention layer in any model is straightforward. We aim to identify a set of hyperparameters that not only maximize attention sparsity but also constrain the attention

error across five different model inputs. To evaluate attention accuracy, we employ a strict error metric, the Relative L1 distance, defined as $L1 = \sum |O - O'| / \sum |O|$. The process begins by setting two L1 error thresholds $l_1$ and $l_2$, e.g., $l_1 = 0.05, l_2 = 0.06$. We first conduct a grid search for $\tau$ and $\theta$ to identify the optimal pair that maximizes sparsity while ensuring $L1 < l_1$. Subsequently, we perform another grid search for $\lambda$ to find the optimal value that further maximizes sparsity while maintaining $L1 < l_2$.

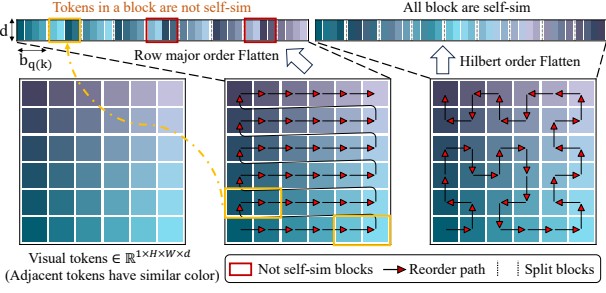

*Figure 5.* Illustration of different token permutation methods in $1 \times 6 \times 6$ space, with block size of 4.

### 3.7. HilbertCurve Permutation

Key idea. Improving sparsity while maintaining accuracy is a key challenge in enhancing the performance of sparse attention. In our algorithm, increasing the self-similarity of key and query blocks can reduce the number of fix blocks. This allows more selective blocks to participate in TopCdf selection, thereby improving sparsity. Since attention is computationally invariant to token permutations, the problem reduces to finding a permutation that enhances the similarity of adjacent tokens.

Image and video models benefit from strong priors: adjacent pixels are likely to be similar. To better leverage this prior, we propose the HilbertCurve permutation, given 3D visual tokens $Q, K, V \in \mathbb{R}^{T \times H \times W \times d}$, We use the Hilbert Curve to fill the 3D space and then flatten tokens along the curve into shape $\mathbb{R}^{L \times d}, L = T \times H \times W$. Fig. 5 illustrates an example of $1 \times 6 \times 6$ visual tokens flattened by row-major order and HilbertCurve. The Hilbert Curve preserves locality effectively, traversing the entire 3D space without crossing rows or columns, thereby increasing the similarity of adjacent tokens and the sparsity of attention.

## 4. Experiment

### 4.1. Setup

**Models.** We validate the effectiveness of SpargeAttn across diverse representative models from language, image, and video generation. Specifically, we conduct experiments on Llama3.1 (8B) (Dubey et al., 2024) for

text-to-text, CogvideoX (2B), Mochi (Team, 2024), and Open-Sora-Plan (Lin et al., 2024) for text-to-video, Flux (.1-dev) (Black Forest Labs, 2023) and Stable-Diffusion3.5 (large) (Stability AI, 2023) for text-to-image.

**Datasets.** The Text-to-text model is evaluated on four zero-shot tasks: WikiText (Merity et al., 2017) to assess the model's prediction confidence, Longbench (Bai et al., 2024) and En.MC of InfiniteBench (Zhang et al., 2024) for a comprehensive assessment of long context understanding capabilities, and the Needle-in-a-Haystack task (Kamradt, 2023) to assess the model's retrieval ability. Text-to-video models are evaluated using the open-sora (Zheng et al., 2024c) prompt sets. Text-to-image models are assessed on COCO annotations (Lin et al., 2014).

**End-to-end metrics.** For Llama3.1, we use perplexity (ppl.) (Jelinek et al., 1977) for WikiText, Longbench score (Bai et al., 2024), and retrival accuracy for the Needle-in-a-Haystack task (Kamradt, 2023). For text-to-video models, following Zhao et al. (2025b), we evaluate the quality of generated videos on five metrics: CLIPSIM and CLIP-Temp (CLIP-T) (Liu et al., 2024b) to measure the text-video alignment; VQA-a and VQA-t to assess the video aesthetic and technical quality, and Flow-score (FScore) for temporal consistency (Wu et al., 2023). For text-to-image models, generated images are compared with the images in the COCO dataset in three aspects: FID (Heusel et al., 2017) for fidelity evaluation, *Clipscore* (CLIP) (Hessel et al., 2021) for text-image alignment, and *ImageReward* (IR) (Xu et al., 2024) for human preference.

**Speed and sparsity metric.** We use TOPS (tera operations per second) to evaluate the speed of sparse attention methods. Specifically, **TOPS** = $O(attn)/t$, where $O(attn)$ represents the total number of operations in a standard attention computation, and $t$ is the latency from a given $(Q, K, V)$ to the output of attention. **Note** that this speed metric is completely fair. This is because the $O(attn)$ is fixed for a set of inputs, and then the speed is determined by $t$, which includes the time spent predicting the sparse region of the attention map. We define **Sparsity** as the proportion of the Matmul of $Q_i K_j^\top$ plus $\widetilde{P}_{ij} V_j$ that are skipped relative to the total number of $Q_i K_j^\top$ plus $\widetilde{P}_{ij} V_j$ in a full attention required.

Implementation and Hyper-parameters. We implement our method using CUDA. As discussed in Sec. 3.6, we need to determine $l_1, l_2$ for models. We use ($l_1 = 0.08, l_2 = 0.09$) for Llama3.1, ($l_1 = 0.05, l_2 = 0.06$) for CogvideoX and Mochi, and ($l_1 = 0.07, l_2 = 0.08$) for Stable-Diffusion3.5 and Flux, ($l_1 = 0.03, l_2 = 0.035$) for Open-Sora-Plan.

**Baselines.** Currently, sparse attention methods applicable

*Table 1.* End-to-end metrics across text, image, and video generation models. ✗ indicates an inability to generate results for evaluation. The speed and sparsity are the average for each layer in the model in real generation tasks described in Sec. 4.1. The speed and sparsity of `Llama3.1` are measured in the Needle-in-a-Haystack task with a 128K sequence length.

| Model (seq_len) | Attention (Sparsity) | Speed (TOPS)↑ | WikiText (Ppl.) ↓ | Longbench ↑ | InfiniteBench ↑ | NIAH ↑ |
|---|---|---|---|---|---|---|
| `Llama3.1` (128K) | Full-Attention | 156.9 | 6.013 | 38.682 | 0.6594 | 0.907 |
| | Minference (0.5) | 140.1 | 10.631 | 28.860 | 0.5152 | 0.832 |
| | FlexPrefill (0.5) | 240.6 | 6.476 | 38.334 | 0.6460 | 0.858 |
| | Minference (0.3) | 115.7 | 6.705 | 34.074 | 0.6532 | 0.870 |
| | FlexPrefill (0.42) | 206.9 | 6.067 | 38.334 | 0.6581 | 0.878 |
| | SpargeAttn (0.54) | **708.1** | **6.020** | **39.058** | **0.6638** | **0.909** |

| Model (seq_len) | Attention (Sparsity) | Speed (TOPS)↑ | CLIPSIM ↑ | CLIP-T ↑ | VQA-a ↑ | VQA-t ↑ | FScore ↑ |
|---|---|---|---|---|---|---|---|
| `CogvideoX` (17K) | Full-Attention | 166.0 | 0.1819 | 0.9976 | 80.384 | 75.946 | 5.342 |
| | Minference (0.5) | 264.6 | 0.1728 | 0.9959 | 70.486 | 62.410 | 2.808 |
| | FlexPrefill (0.6) | 175.3 | 0.1523 | 0.9926 | 1.5171 | 4.5034 | 1.652 |
| | Minference (0.3) | 196.9 | 0.1754 | 0.9964 | 77.326 | 63.525 | 3.742 |
| | FlexPrefill (0.45) | 142.0 | 0.1564 | 0.9917 | 7.7259 | 8.8426 | 2.089 |
| | SpargeAttn (0.46) | **507.9** | **0.1798** | **0.9974** | **78.276** | **74.846** | **5.030** |
| `Mochi` (22K) | Full-Attention | 164.2 | 0.1725 | 0.9990 | 56.472 | 67.663 | 1.681 |
| | Minference (0.5) | 202.4 | 0.1629 | 0.9891 | 6.668 | 50.839 | 0.653 |
| | FlexPrefill (0.48) | 191.3 | 0.1667 | 0.9898 | 0.582 | 0.0043 | ✗ |
| | Minference (0.3) | 147.7 | 0.1682 | 0.9889 | 14.541 | 42.956 | 0.833 |
| | FlexPrefill (0.4) | 171.7 | 0.1677 | 0.9909 | 2.941 | 0.7413 | ✗ |
| | SpargeAttn (0.47) | **582.4** | **0.1720** | **0.9990** | **54.179** | **67.219** | **1.807** |

| Model (seq_len) | Attention (Sparsity) | CLIPSIM ↑ | CLIP-T ↑ | VQA-a ↑ | VQA-t ↑ | FScore ↑ | Latency ↓ |
|---|---|---|---|---|---|---|---|
| `Open-Sora-Plan` (38K) | Full-Attention | 0.1650 | 0.9994 | 81.40 | 80.60 | 0.847 | 629s |
| | SpargeAttn (0.34) | 0.1686 | 0.9985 | 77.59 | 76.91 | 0.839 | **393s** |

| Model (seq_len) | Attention (Sparsity) | Speed (TOPS)↑ | FID ↓ | CLIP ↑ | IR ↑ |
|---|---|---|---|---|---|
| `Flux` (4.5K) | Full-Attention | 158.2 | 166.103 | 31.217 | 0.8701 |
| | Minference (0.5) | 151.8 | 180.650 | 30.235 | 0.4084 |
| | FlexPrefill (0.48) | 47.7 | 443.928 | 18.3377 | -2.2657 |
| | Minference (0.3) | 118.9 | 170.221 | 31.001 | 0.7701 |
| | FlexPrefill (0.41) | 40.9 | 405.043 | 19.5591 | -2.2362 |
| | SpargeAttn (0.38) | **280.3** | **163.982** | **31.448** | **0.9207** |
| `Stable-Diffusion3.5` (4.5K) | Full-Attention | 164.2 | 166.101 | 32.007 | 0.9699 |
| | Minference (0.5) | 186.4 | 348.930 | 18.3024 | -2.2678 |
| | FlexPrefill (0.37) | 23.1 | 350.497 | 18.447 | -2.2774 |
| | Minference (0.3) | 150.3 | 337.530 | 18.099 | -2.2647 |
| | FlexPrefill (0.35) | 22.7 | 348.612 | 18.147 | -2.2756 |
| | SpargeAttn (0.31) | **293.0** | **166.193** | **32.114** | **0.9727** |

across different model types are limited. We choose block-sparse MInference (Jiang et al., 2024) and FlexPrefill (Lai et al., 2025) as our baselines. To vary the *sparsity* of these baselines, we use 30% and 70% for MInference, and use $\gamma = 0.95$ and $0.99$ for FlexPrefill according to their paper.

## 4.2. Quality and Efficiency Evaluation

**End-to-end metrics.** We assess the end-to-end metrics of various models using `SpargeAttn` compared to using full attention and baselines. Table 1 shows the results. We can observe that our method incurs almost no end-to-end metric loss across various models compared to Full-Attention and surpasses baselines with various sparsity levels in terms of end-to-end accuracy. Fig. 6, 7, 8, and 12 show some visible comparison examples on `CogvideoX`,

`Flux`, `Stable-Diffusion3.5`, `Mochi`, and `Open-Sora-Plan`, showing that `SpargeAttn` incurs no performance loss and outperforms baselines.

**Attention speed.** Table 1 shows that our method achieves faster speeds compared to Full-Attention and surpasses baselines with various sparsity levels in terms of attention speed. Fig. 10 illustrates the kernel speeds of various methods across different sparsity, highlighting the efficiency of our approach and its significant advantage over other methods.

## 4.3. Ablation Study and key Insights

**Overhead of sparse block prediction.** Table 3 compares the overhead of dynamic sparse block prediction in `SpargeAttn` compared with attention execution latency.

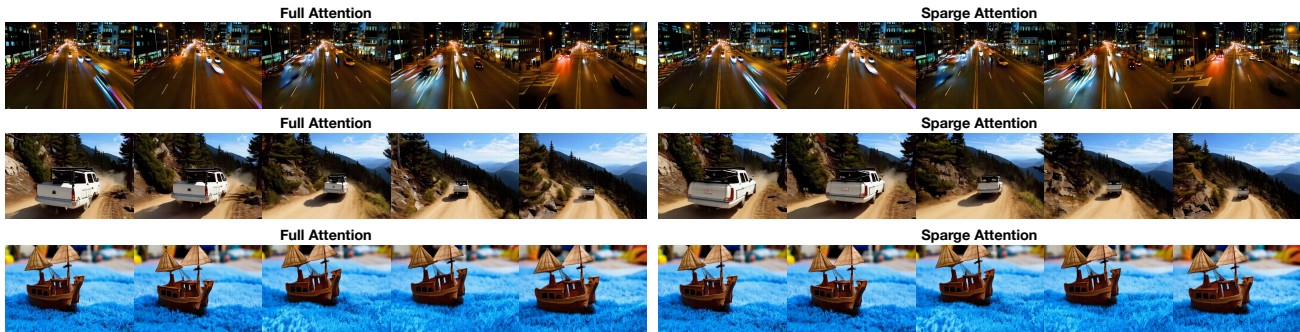

*Figure 6.* Visible examples on CogvideoX using SpargeAttention.

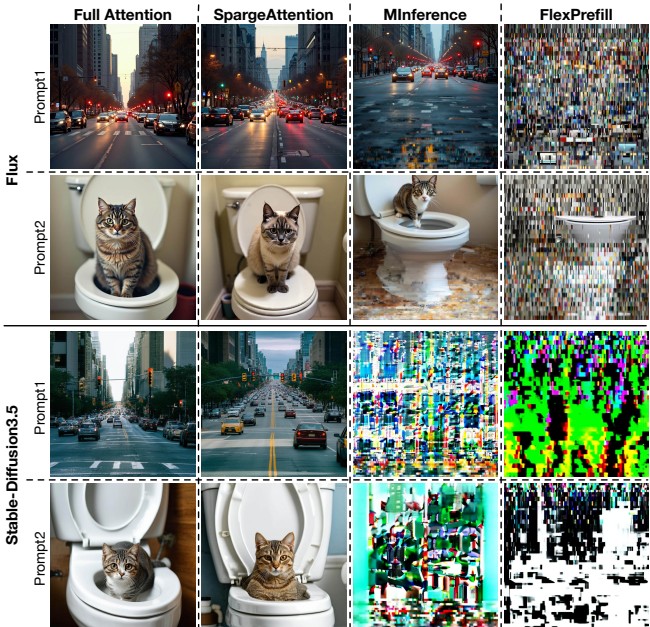

*Figure 7.* Comparison examples on `Flux` and `Stable-Diffusion3.5`. The sparsity of SpargeAttn, MInference and FlexPrefill is 0.38, 0.3, and 0.4 on `Flux` and 0.31, 0.3, and 0.35 on `Stable-Diffusion3.5`.

The results indicate that the prediction overhead is minimal compared to attention, particularly for longer sequences.

*Table 2.* End-to-end generation latency using `SpargeAttn`.

| Model | GPU | Original | SageAttn | SpargeAttn |
|---|---|---|---|---|
| CogvideoX | RTX4090 | 87 s | 68 s | **53 s** |
| Mochi | L40 | 1897 s | 1544 s | **1037 s** |
| Llama3.1 (24K) | RTX4090 | 4.01 s | 3.53 s | **2.6 s** |
| Llama3.1 (128K) | L40 | 52 s | 42s | **29.98 s** |

**End-to-end speedup.** Table 2 shows the end-to-end latency on `CogvideoX`, `Mochi`, and `Llama3.1` using `SpargeAttn`. Notably, `SpargeAttn` achieves 1.83x speedup on `Mochi`.

**Effect of Hilbert Curve permutation.** We evaluate the

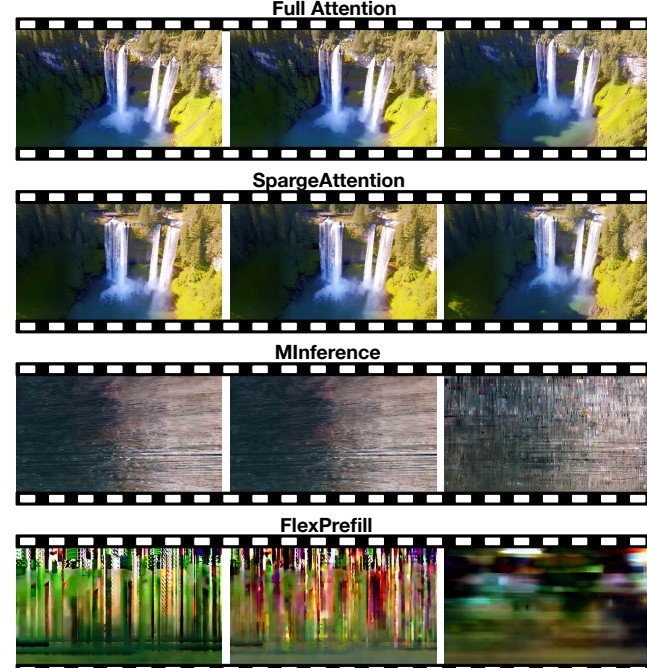

*Figure 8.* Comparison examples on `Mochi`. The sparsity of `SpargeAttn`, MInference and FlexPrefill is 0.47, 0.3, and 0.4.

*Table 3.* Overhead of sparse block prediction in `SpargeAttn`.

| Sequence Len | Prediction (ms) | Full Attention (ms) | Overhead |
|---|---|---|---|
| 8k | **0.251** | 6.649 | 3.78% |
| 16k | **0.487** | 26.83 | 1.82% |
| 32k | **0.972** | 106.68 | 0.911% |
| 64k | **2.599** | 424.24 | 0.612% |
| 128k | **8.764** | 1696.2 | 0.516% |

impact of Hilbert Curve permutation on `Mochi` by comparing three metrics: average block similarity across blocks of query or key, L1 error defined in Sec. 3.6, and *sparsity*. Table 4 shows that the HilbertCurve permutation consistently achieves superior block self-similarity and sparsity, with only a marginal difference in accuracy. Please see Appendix A.1 for more analysis and details.

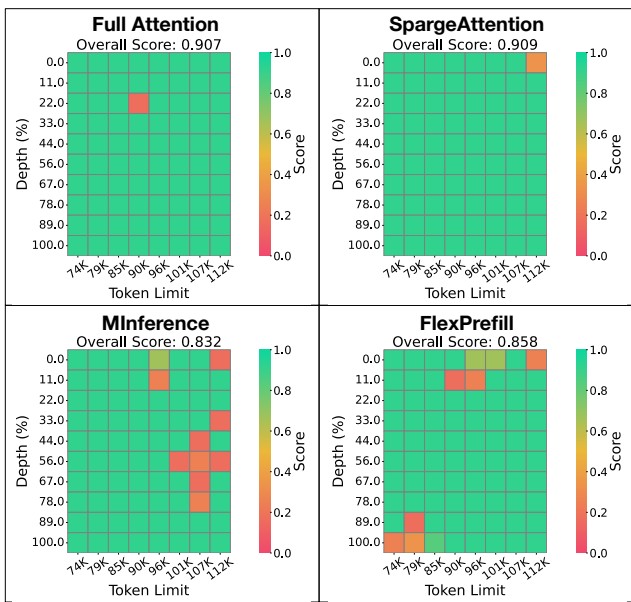

*Figure 9.* A Needle-in-a-Haystack comparison example on `Llama3.1`. The sparsity of `SpargeAttn`, MInference, and FlexPrefill is 0.5, 0.5, and 0.54.

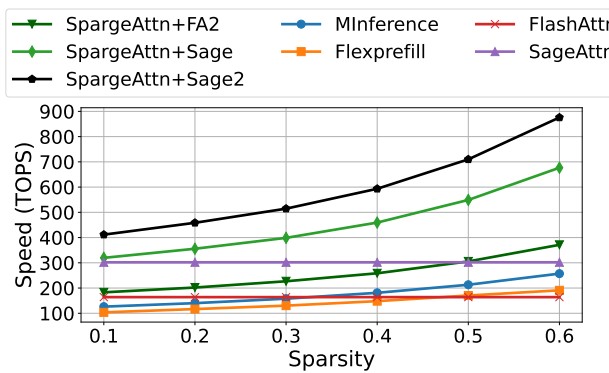

*Figure 10.* Kernel speed comparison under varying sparsity. Input tensors have a sequence length of 22K and a head dimension of 128. *SpargeAttn+FA2* means deploying our method on FlashAttention2.

*Table 4.* Effect of permutation on sparsity and accuracy. Sim-q and Sim-k are the average block self-similarity of the query and key.

| Method | Sim-q ↑ | Sim-k ↑ | L1 ↓ | Sparsity ↑ |
|---|---|---|---|---|
| Random | 0.321 | 0.019 | 0.0414 | 0.048 |
| Rowmajor | 0.551 | 0.390 | **0.0307** | 0.363 |
| Timemajor | 0.514 | 0.367 | 0.0342 | 0.338 |
| HilbertCurve | **0.572** | **0.479** | 0.0389 | **0.392** |

*Table 5.* Abalation of self-similarity judge.

| Method | VQA-a ↑ | VQA-t ↑ | FScore ↑ |
|---|---|---|---|
| W/o. self-sim Judge | 34.664 | 44.722 | 1.138 |
| With self-sim Judge | 54.179 | 67.219 | 1.807 |

*Table 6.* Analysis of sparsity from $M_g$ and $M_{pv}$.

| Strategy | only $M_g$ | only $M_{pv}$ | $M_g + M_{pv}$ |
|---|---|---|---|
| Sparsity | 51.2% | 27.7% | 54% |

**Ablation of self-similarity judge** We ablate the effect of the self-similarity judge on `Mochi`. As shown in Table 5, we find that self-similarity judge can guarantee end-to-end accuracy. Please see Appendix A.2 for more analysis.

**Analysis of sparsity from $M_g$ and $M_{pv}$.** Table 6 shows the sparsity when only using $M_g$, only using $M_{pv}$, and using $M_g + M_{pv}$ on `Llama3.1` in Needle-in-a-Haystack task with 128K sequence length.

**SpargeAttn enhance the LLM performance.** From Table 1, Fig. 9 and 11, we observe that `SpargeAttn` enhances LLM performance in long-context tasks. This improvement may result from the fact that sparse attention helps the LLM focus on more relevant information.

*Table 7.* Sparsity increases with sequence length under a constant accuracy bound on `Llama3.1`.

| Sequence Len | 8K | 16K | 24K | 48K | 128K |
|---|---|---|---|---|---|
| **Sparsity** | 6.8% | 26.4% | 35.7% | 49.8% | 54% |

**Sparsity increases with sequence length.** As shown in Table 7, we find that on `Llama3.1`, sparsity increases with sequence length. This suggests that the longer contexts, the higher speedup of `SpargeAttn` can achieve.

**Sparsity analysis over diffusion model.** We conduct a detailed analysis of sparsity in `CogvideoX` across all layers, heads, timesteps, and samples using `SpargeAttn` to get more insights (See Appendix A.4 for detailed figures). We find that sparsity varied with layers and heads, indicating that setting different hyperparameters for each layer and head is necessary. We also find that for diffusion models, the sparsity increases with the sample timesteps.

## 5. Conclusion

In this paper, we propose `SpargeAttn`, a universal sparse and quantized attention that executes attention efficiently and accurately for any input. Our method uses a two-stage online filter: in the first stage, we rapidly and accurately predict the attention map, enabling the skip of some matrix multiplications in attention. In the second stage, we design an online softmax-aware filter that incurs no extra overhead and further skips some matrix multiplications. Experiments show that `SpargeAttn` accelerates diverse models, including language, image, and video generation models, without sacrificing end-to-end metrics.

## Acknowledgment

This work was supported by the NSFC Projects (Nos. 92270001, 62376131). J.Z is also supported by the XPlorer Prize.

## Impact Statement

This paper presents work that aims to advance the field of Machine Learning. There are many potential societal consequences of our work, none of which we feel must be specifically highlighted here.

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

# A. Appendix

## A.1. Detailed Explain and results of permutation ablation

We use five distinct prompts and pre-searched hyperparameters with $l_1 = 0.05, l_2 = 0.06$ on both `CogvideoX` and `Mochi` models. The permutation are performed separately in attention operation for $Q, K, V$ after position embedding. To retain the original order of the input sequence, an inverse permutation is performed on the output of attention; for models using visual-language joint self-attention(e.g., `CogvideoX`), we only permute the visual tokens. When evaluating block self-similarity, we choose a block size of $128$ for query and $64$ for key, which aligns with our kernel implementation. The precision metric(L1) is evaluated using FlashAttention2 output as ground truth.

We choose different permutation methods to compare their impact on the performance of attention operations. Given a 3D visual token tensor with shape $T \times H \times W \times d$, the permutation finally results in a tensor with shape $L \times d$, where $L = T \times H \times W$. The permutation methods and their detailed descriptions are shown in Table 8.

*Table 8.* The detailed description of different permutation methods.

| Method | Detailed Description |
|---|---|
| Random | Random permutation of tokens, the order is recorded to perform inverse permutation. |
| Rowmajor | Permutation following row-major order. Tokens are continuous along the W dimension. |
| Columnmajor | Permutation following column-major order. Tokens are continuous along the H dimension. |
| Timemajor | Permutation following time-major order. Tokens are continuous along the T dimension. |
| HilbertCurve | Permutation following a Hilbert curve. |

Detailed results of permutation ablation for the `CogvideoX` and `Mochi` models are presented in Table 9. The HilbertCurve permutation consistently achieves superior block self-similarity and sparsity, with only a marginal loss in precision. This suggests that the HilbertCurve permutation effectively enhances block self-similarity and sparsity. It is worth noting that the random permutation retains the precision metrics but sacrifices sparsity. This indicates that our algorithm has the property of dynamically adjusting and robust to complex token sequences.

*Table 9.* The impact of permutation on `CogvideoX` and `Mochi` models. Sim-q is the block self-similarity of the query, and Sim-k is the block self-similarity of the key.

| Method | Sim-q↑ | | Sim-k↑ | | Precision(L1)↓ | | Sparsity↑ | |
|---|---|---|---|---|---|---|---|---|
| | CogvideoX | Mochi | CogvideoX | Mochi | CogvideoX | Mochi | CogvideoX | Mochi |
| Random | 0.502 | 0.321 | 0.025 | 0.019 | 0.0348 | 0.0414 | 0.027 | 0.048 |
| Rowmajor | 0.676 | 0.551 | 0.435 | 0.390 | **0.0265** | **0.0307** | 0.242 | 0.363 |
| Columnmajor | 0.633 | 0.547 | 0.335 | 0.394 | 0.0274 | 0.0342 | 0.198 | 0.366 |
| Timemajor | 0.692 | 0.514 | 0.479 | 0.367 | 0.0294 | 0.0342 | 0.238 | 0.338 |
| HilbertCurve | **0.709** | **0.572** | **0.523** | **0.479** | 0.0323 | 0.0389 | **0.265** | **0.392** |

## A.2. Ablation Study of Self-Similarity Judge

To investigate the impact of the self-similarity judge on attention performance, we follow the experimental setting outlined in Sec. A.1 and conduct an ablation study by removing the self-similarity judge. In most cases, the presence of highly localized patterns results in a minimal number of non-self-similar blocks, leading to only minor differences in precision and sparsity when averaging across all tensor cases. To obtain more meaningful and interpretable insights, we specifically analyze cases where the precision difference is statistically significant.

To this end, we apply a threshold-based selection criterion, retaining only those cases where the absolute difference between $L1^{sim-judge}$ (precision error with the self-similarity judge) and $L1^{no-judge}$ (precision error without the self-similarity judge) exceeds 0.05. This criterion results in approximately 2% of the tensor cases being retained for further analysis. We employ precision (L1 error) and sparsity as evaluation metrics to assess the influence of the self-similarity judge on the attention output. The results are summarized in Table 10.

The findings demonstrate that the self-similarity judge effectively mitigates extreme precision loss while introducing only a marginal reduction in sparsity. Furthermore, we observe that a significant proportion of cases exhibiting notable differences

*Table 10.* Impact of the self-similarity judge on the accuracy and sparsity of attention.

| Method | w/ judge | | w/o judge | | filter w/ judge | | filter w/o judge | |
|---|---|---|---|---|---|---|---|---|
| | CogvideoX | Mochi | CogvideoX | Mochi | CogvideoX | Mochi | CogvideoX | Mochi |
| L1 error↓ | 0.0316 | 0.0343 | 0.0325 | 0.0365 | 0.0843 | 0.0555 | 0.214 | 0.154 |
| Sparsity ↑ | 0.199 | 0.301 | 0.203 | 0.305 | 0.242 | 0.371 | 0.275 | 0.392 |

originate from the Random permutation category in the `CogvideoX` model. This observation further highlights the role of the self-similarity judge in enhancing the model's robustness to complex token sequences while maintaining high precision.

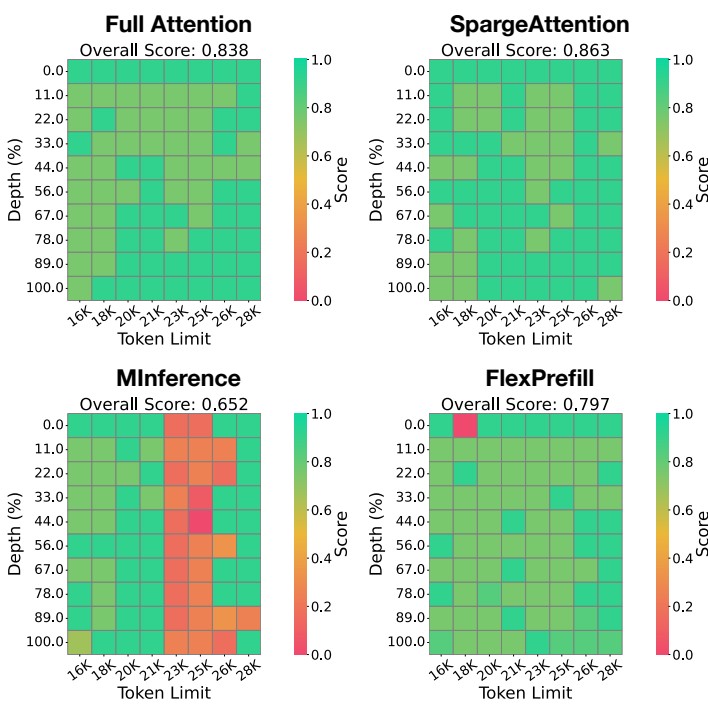

*Figure 11.* A Needle-in-a-Haystack comparison example on `Llama3.1`. The sparsity of `SpargeAttn`, MInference, and FlexPrefill is 0.36, 0.3, and 0.3.

*Table 11.* End-to-end metrics on `Llama3.1` in the Needle-in-a-Haystack task with 16-28K sequence lengths.

| **Model** (seq_len) | **Attention** (Sparsity) | **Speed (TOPS)**↑ | **NIAH** ↑ |
|---|---|---|---|
| | Full-Attention | 156.9 | 0.838 |
| | Minference (0.5) | 122.5 | 0.635 |
| Llama3.1 | FlexPrefill (0.6) | 179.6 | 0.776 |
| (24K) | Minference (0.3) | 102.3 | 0.652 |
| | FlexPrefill (0.3) | 117.6 | 0.797 |
| | SpargeAttn (0.36) | **443.6** | **0.863** |

## A.3. Additional Experiments

In this section, we present additional experimental results further to evaluate the performance of `SpargeAttn` compared to baselines. Fig. 11 and 11 show the results on `Llama3.1` in the Needle-in-a-Haystack task with 16-28K sequence length. Fig 13 shows a visible comparison example on `Mochi`.

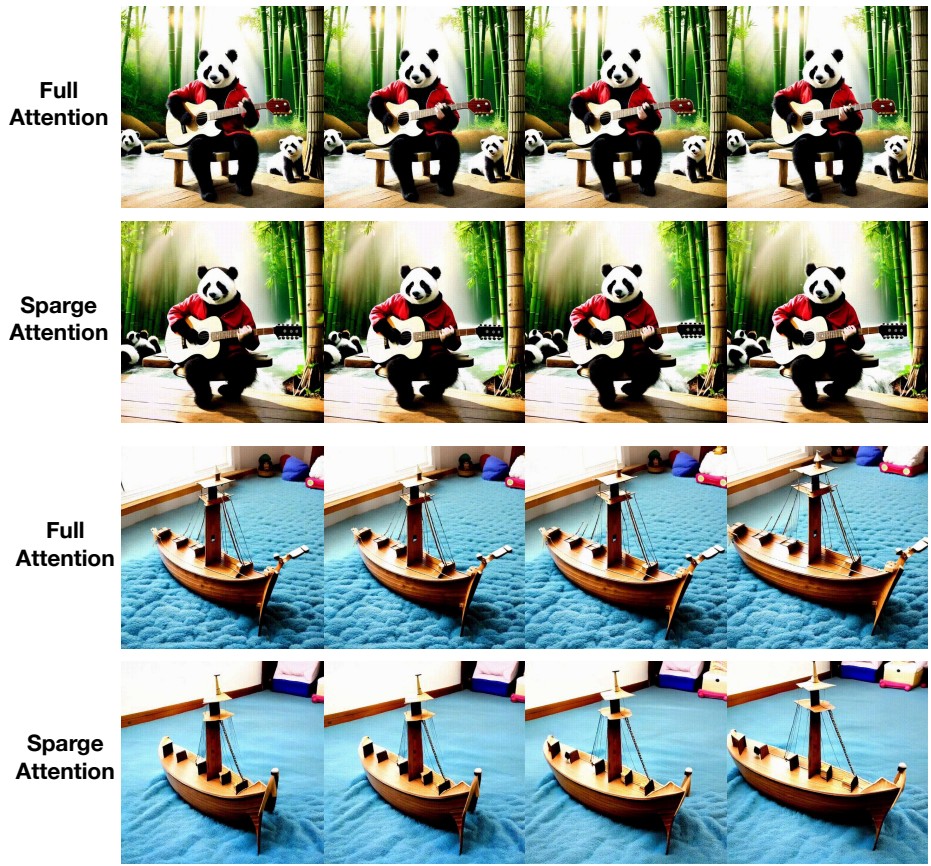

*Figure 12.* Visible examples on Open-sora-Plan.

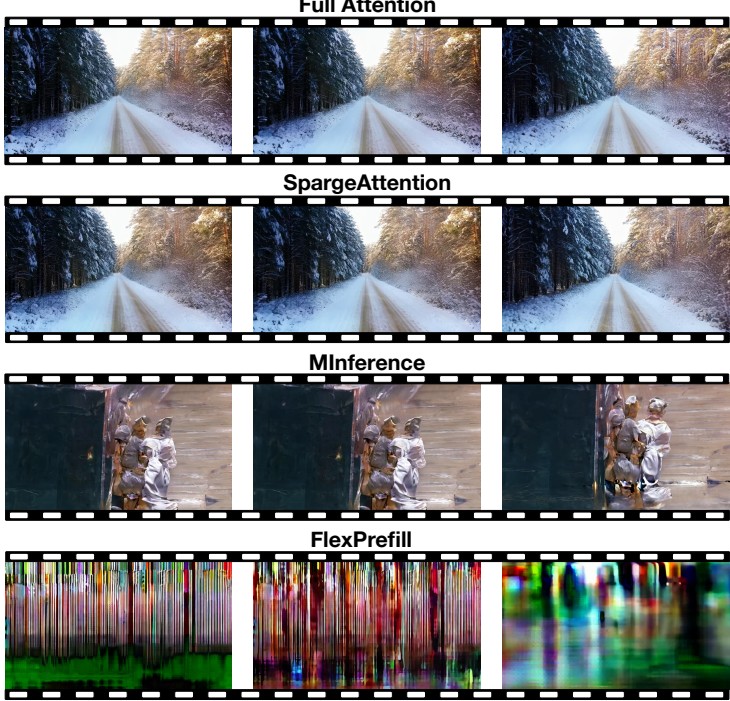

*Figure 13.* Comparison examples on `Mochi`. The sparsity of `SpargeAttn`, MInference, and FlexPrefill is 0.47, 0.3, and 0.4.

## A.4. Sparsity analysis over diffusion model

In this section, we analyze the sparsity patterns in `CogvideoX` across different dimensions: model layers, denoising timesteps, input samples, and attention heads. Figure 14 illustrates the layer-wise sparsity. Figure 15 demonstrates timestep-wise sparsity. Figure 16 highlights sample-wise sparsity. Figure 17 presents head-wise sparsity, illustrating the diversity in attention behavior across different heads. These analyses are helpful for the design of some diffusion algorithms (Zheng et al., 2023; 2024b;a; 2025; Zhao et al., 2024; 2025a; Wang et al., 2024).

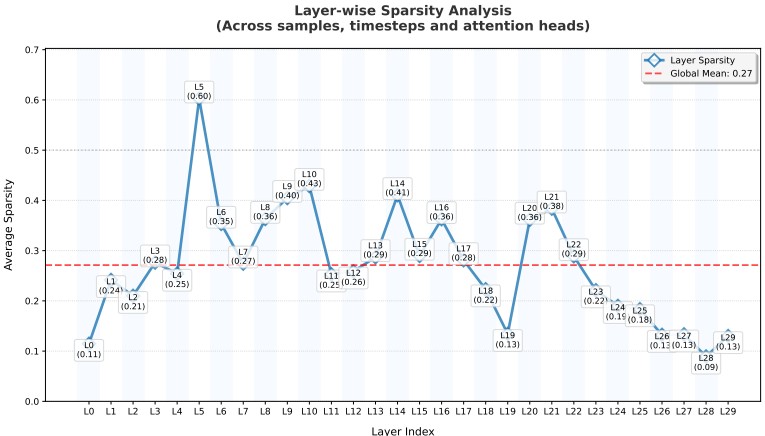

*Figure 14.* Layer-wise sparsity of `CogvideoX`.

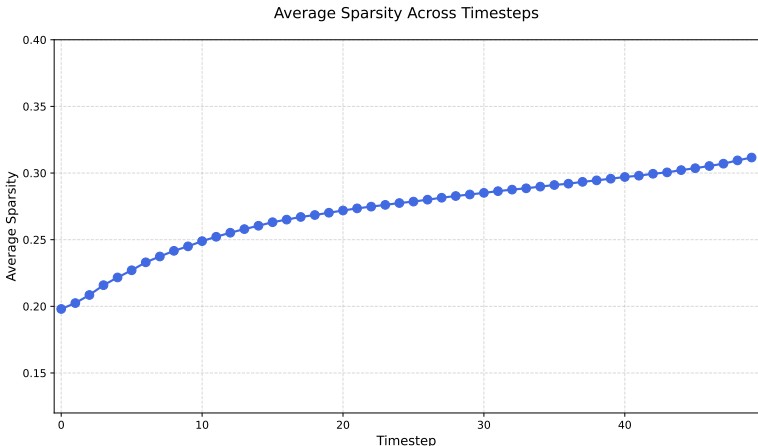

*Figure 15.* Timestep-wise sparsity of `CogvideoX`.

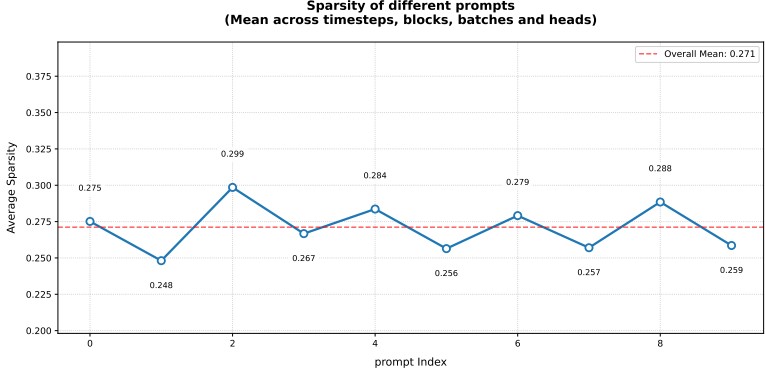

*Figure 16.* Sample-wise sparsity of `CogvideoX`.

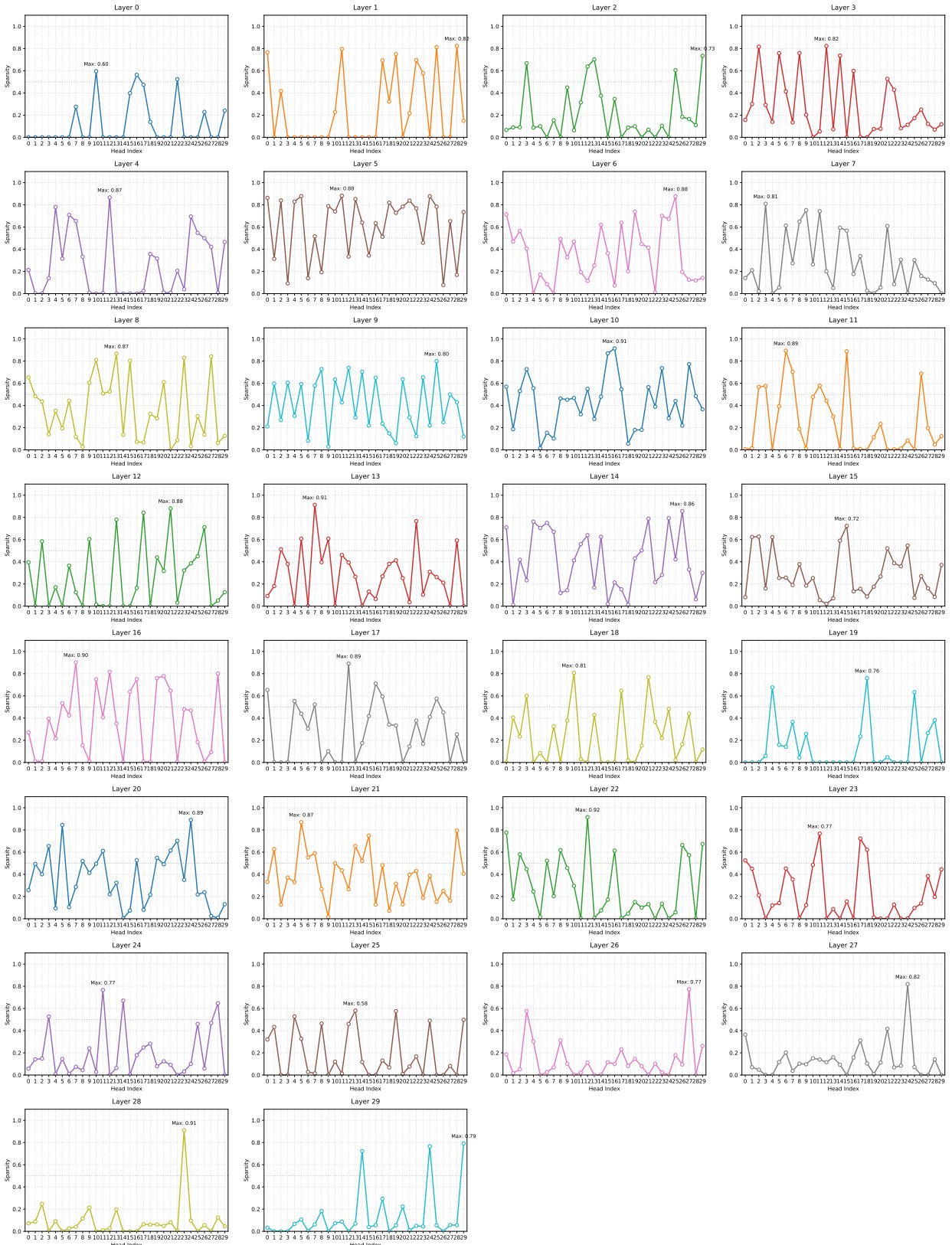

*Figure 17.* Head-wise sparsity of `CogvideoX`.

