# OpenReview forum: "SpargeAttention: Accurate and Training-free Sparse Attention Accelerating Any Model Inference"
_ICML.cc/2025/Conference — ICML 2025 poster_

### Official Review · Reviewer_YtrB · 2025-03-12

**Overall Recommendation:** 4

**Summary:**

The authors propose a method for sparse attention computation which works for both language and visual models. The method constructs a sparse mask using mean pooling of blocks of queries and keys along with a measure of self similarity within the blocks.

**Claims And Evidence:**

The experiments seem to validate the method, which shows strong performance on both language and visual tasks.

**Essential References Not Discussed:**

The references covered are sufficient.

**Experimental Designs Or Analyses:**

The experimental design appears to be sound.

**Methods And Evaluation Criteria:**

The evaluations are relevant to the method.

**Other Comments Or Suggestions:**

I would suggest rewriting the method section to have more intuitive explanations for the derived method. To create space, the precise algorithm and a few equations could be moved to the appendix.

**Other Strengths And Weaknesses:**

## Weaknesses

I find the presentation of the method confusing to follow. In particular, I do not think the explanation of the self similarity was properly motivated or explained.

- I am confused by this statement --> "Importantly, compressing only the token blocks with high self-similarity is crucial, as omitting computations for non-self-similar blocks can result in the loss of critical information. This will be confirmed in Sec. 4 and A.2."
  - If you compress only token blocks with high self similarity, it sounds as if you are omitting computations for non-self similar blocks by definition.

- I am also confused by this statement --> "Finally, we need to ensure that calculations involving nonself-similar blocks of Q or K are not omitted." The algorithm described up until this point went to great lengths to eliminate blocks which are not self similar. If you want to ultimately include these blocks regardless, then why do we need to eliminate them?

- I do not understand the significance of section 3.7 which describes the HilbertCurve Permutation. While I can see that comparing neighboring pixels in this way would be advantageous for image models, I do not get how this applies to the self similarity which is proposed by the algorithm thus far.

  - To my knowledge, the self similarity described in L213C1 should be like $Q_i \in \mathbb{R}^{b_q \times d}$ and the cosine similarity would be calculated as something like $\sum \frac{Q_i Q_i^\top}{\vert max(Q_i Q_i^\top) \vert}$. I assume the sum because until this point, there has been no mention of how this $d_q \times d_q$ matrix is reduced to a scalar.
  - Therefore, if the reduction is a permutation invariant sum, then what benefit could the HilbertCurve Permutation provide?

**Questions For Authors:**

Please see the above "Weaknesses" section. I do believe the authors can clear up any misconceptions during the rebuttal, so I await their responses.

## Post Rebuttal

Thank you for answering my questions in my review. You have cleared up many of my concerns. I have raised my score.

**Relation To Broader Scientific Literature:**

The contributions are relevant to recently published literature in sparse attention.

**Theoretical Claims:**

The correctness of the algorithm seems to be correct, however, there are confusing parts which are not well specified in my opinion. See further comments for details.

---

> ### Author Rebuttal · Authors · 2025-03-31
>
> Dear Reviewer YtrB,
> Thank you for your valuable questions. Below, we address each point raised.
>
> ---
> >### Weaknesses1
>
> **Reply**: Sorry for the confusion. To clarify: We need to predict sparse areas in the attention map to skip unnecessary computations. However, a naive approach - compressing all blocks of $Q, K$ via mean pooling to get a compressed attention map $P$ - is inaccurate, because mean pooling cannot properly represent non-self-similar blocks. Therefore, we only apply mean pooling to self-similar blocks to generate the compressed attention map. The non-self-similar blocks skip this prediction phase and directly participate in the full attention computation.
>
> Thank you for your suggestion. We will revise our paper to make the method part be clear.
>
> ---
> >### Weaknesses1.1
>
> **Reply**: There may be some misunderstanding between compressing and computation. Not compressing does not mean not computation. Compressing blocks is for judging which blocks can be omitted from attention computation. We clarify the concepts of compressing, computation, self-similar blocks, and non-self-similar blocks:
>
> - *Computation*: The FlashAttention computation, i.e., the block matrix multiplications of $QK^T$ and $PV$.
>
> - *Compressing*: Compressing is doing mean pooling for blocks of $Q, K$ to judge which blocks of $Q, K, V$ can be omitted from attention computation. Because the mean pooling results of non-self-similar blocks can not represent the information of blocks, we skip the compressing process for non-self-similar blocks, but directly compute all of them in attention, i.e., we do not omit any computation for Non-self-similar blocks.
>
> We will revise the paper to rename the 'self-similar blocks' to 'selective blocks' and the 'non-self-similar blocks' to 'fix blocks'.
>
> ---
> >### Weaknesses2
>
> **Reply**: Sorry for the confusion. We do not use the word 'eliminate' in our paper, and we suppose the 'eliminate' means not participating in the compressing. Therefore, as explained in the reply for *W1.1*, non-self-similar will not be compressed, but directly participate in the attention computation.
>
> ---
> >### Weaknesses3
>
> **Reply**: The embeddings of tokens corresponding to similar pixels are relatively similar. The HilbertCurve clusters tokens from similar pixels together, increasing the similarity between the blocks of Q and K. As a result, more 'selective blocks' participate in the sparse prediction process, while the 'fixed blocks' that always computed decrease, leading to higher sparsity.
>
> ---
> >### Weaknesses3.1
>
> **Reply**: Thank you for pointing out the typo in the equation. Actually, we will compute a mean of the dq×dq matrix as a value, just as Line 201 of our paper: "We first compute a mean cosine similarity across tokens for each block of Q and K."
>
> ---
> >### Weaknesses3.2
>
> **Reply**: Yes, standard attention is token permutation invariant. However, we use sparse attention. After the HilbertCurve permutation, the similarity in blocks of Q, K increases, which raises attention sparsity. This allows us to omit more computations, improving speed. As shown in Table 9, HilbertCurve Permutation increases block self-similarity, raising the sparsity of attention.
>
> An Example:
>
> *Without permutation:*
> - Self-similar blocks: 80% (sparsity within similar blocks=0.3)
> - Non-similar blocks: 20% (always computed in attention)
> *Effective sparsity*: 0.3×0.8=0.24
> *Blocks computed*: (1-0.3)×0.8+0.2=0.76
>
> *With permutation:*
> - Self-similar blocks: 90% (sparsity within similar blocks=0.3)
> - Non-similar blocks: 10% (always computed in attention)
> *Effective sparsity*: 0.3×0.9=0.27
> *Blocks computed*: (1-0.3)×0.9+0.1=0.73
>
> ---
> >Last but importantly, the Hilbertcurve permutation is a relatively minor aspect of our work. The **key contributions** of SpargeAttn are:
>
> 1. **Effectiveness**: We design the first sparse attention method that can actually accelerate **across language, image, and video models** without compromising accuracy.
>
> 2. **Method Innovations**:
> - First to enable block-wise sparse computation via selective compression
> - First to propose sparse online softmax (a fundamentally novel approach)
> - First to establish guaranteed error bounds for all attention layers in the model
>
> We summarize some representative methods from four aspects, including whether needing a training process, whether relying on specific attention map patterns, whether being applicable to all models (language and diffusion), and whether implementing attention quantization:
>
> |Method|Training Free|Pattern Free|Universal|Quantization|
> |-|-|-|-|-|
> |MInference|✓|✓|✓|-|
> |DuoAttention|✗|✗|✗|-|
> |SeerAttention|✗|✓|✗|-|
> |FlexPrefill|✓|✓|✓|-|
> |H2O|✓|✗|✗|-|
> |InfLLM|✓|✗|✗|-|
> |DitFastAttn|✓|✗|✗|-|
> |SparQAttn|✓|✓|✗|-|
> |LokiAttn|✓|✓|✗|-|
> |SampleAttention|✓|✗|✗|-|
> |FastAttention|✗|✓|✗|-|
> |MOA|✗|✗|✗|-|
> |Reformer|✗|✓|✓|-|
> |**Ours**|**✓**|**✓**|**✓**|**✓**|
>
> ---
> ---
> If your concerns have been resolved, we would greatly appreciate it if you consider raising the score.

---

### Official Review · Reviewer_m9qG · 2025-03-14

**Overall Recommendation:** 3

**Summary:**

The paper proposes SpargeAttn, a universal sparse and quantized attention for any model, accelerates diverse models, including language, image, and video generation, without sacrificing end-to-end metrics. For blocks composed of highly similar tokens, they consolidate these tokens into a single representative token for the block, skipping the computation and further identify the small enough values in the attention map during the online softmax process.

**Claims And Evidence:**

Yes

**Essential References Not Discussed:**

please check STA (FastVideo) whether they are  concurrent work. If not, it would be better to compare with them.

**Experimental Designs Or Analyses:**

Yes

**Methods And Evaluation Criteria:**

Yes.
1. Lack in generated video / video demos as the metrics evaluation only is not reliable
2. Lack in report of Attention score recall and l2 difference of output video/image (given same initial noise with fixed seed)

**Other Comments Or Suggestions:**

1. Provide with more analysis on sparsity of each head.
2. Provide more conclusive description on main difference to previous methods and core contribution to community.
3. Provide video demos

**Other Strengths And Weaknesses:**

Weakness:
1. Lack in generated video / video demos as the metrics evaluation only is not reliable
1. Lack in analysis on sparsity of each head
2. Lack in generalization to multi-GPU parallel inference settings

**Questions For Authors:**

1. Current evaluation (FID, Clip score, etc.) is not reliable. Please provide attention score recall and l2 difference of output video/image if given same initial noise with fixed seed.
2. While the proposed method demonstrates promising results on a single GPU, it would be valuable to further explore how the approach handles potential load imbalance issues in sequence and head parallelism scenarios, particularly given the varying sparsity rates across different heads. Additionally, an investigation into whether the acceleration benefits can be maintained in multi-GPU parallel inference settings would significantly strengthen the practical relevance of the work. These aspects could provide interesting directions for future research and further enhance the applicability of the method in real-world scenarios.
3. The paper presents a universal approach for online acceleration. However, it would be insightful to investigate whether certain heads might pose challenges to acceleration due to their inherent characteristics. Specifically, providing a detailed analysis of the sparsity levels and acceleration performance for each individual head would strengthen the study. This analysis could help identify potential limitations or exceptions to the universality claim and offer a more comprehensive understanding of the method's applicability across different scenarios.
4. The authors emphasize the universality of their proposed acceleration method as a key contribution. However, it would be valuable to explore whether the method remains effective when applied to models that have already incorporated sparse architectural designs during the pre-training phase. For instance, demonstrating the acceleration performance on state-of-the-art open-source models, such as the latest version of Opensora-plan, could provide compelling evidence of the method's robustness and generalizability across diverse model architectures. This additional analysis would further strengthen the universality claim and enhance the practical relevance of the work.
5. analysis on different timesteps of diffusion video model

I will raise my score if most of my concerns could be addressed.

**Relation To Broader Scientific Literature:**

a new pattern-free, training-free attention accleration method for video generation due to its well generalization

**Theoretical Claims:**

No issues

---

> ### Author Rebuttal · Authors · 2025-03-31
>
> Dear Reviewer m9qG,
> Thank you for your valuable suggestions and questions.
>
> ---
> >### Essential References
>
> **Reply**: We check STA (FastVideo) and confirm it is after the ICML submission deadline.
>
> ---
> >### Comment1 and Question5
>
> **Reply**: Thank you for your valuable suggestion. We conducted detailed analysis and visualization of sparsity and sparse patterns on CogVideoX across all layers, timesteps, and heads at https://anonymous.4open.science/r/SpargeAttn_Re, named **[Analysis Repo]**.
>
> According to **[Analysis Repo]**, we conclude that:
>
> **Analysis of heads**: (1) There is a noticeable variation in sparsity across different heads. (2) Different heads exhibit distinct sparsity patterns.
>
> **Analysis of timesteps**: Sparsity increases as the sampling steps. This aligns with the intuition that image noise diminishes, allowing more semantic patterns to emerge and highlighting the advantage of the dynamic sparse method.
>
> **Analysis of Layers**: The model exhibits lower sparsity in the initial and final layers, while intermediate layers tend to have higher sparsity.
>
> ---
> >### Comment2
>
> **Reply**: Thank you for the suggestion, and we summarize our contributions:
>
> **Key Contributions:**
>
> 1. **Effectiveness**: We design the first sparse attention method that can actually accelerate **across language, image, and video models** without compromising accuracy.
>
> 2. **Method Innovations**:
> - First to enable block-wise sparse computation via selective compression.
> - First to propose sparse online softmax (a fundamentally novel approach).
> - First to establish guaranteed error bounds for all attention layers in the model.
>
> Moreover, we summarize some representative methods from four aspects, including whether needing a training process, whether relying on specific attention map patterns, whether being applicable to all models (language and diffusion), and whether implementing attention quantization:
>
> |Method|Training Free|Pattern Free|Universal|Quantization|
> |-|-|-|-|-|
> |MInference|✓|✓|✓|-|
> |DuoAttention|✗|✗|✗|-|
> |SeerAttention|✗|✓|✗|-|
> |FlexPrefill|✓|✓|✓|-|
> |H2O|✓|✗|✗|-|
> |InfLLM|✓|✗|✗|-|
> |DitFastAttn|✓|✗|✗|-|
> |SparQAttn|✓|✓|✗|-|
> |LokiAttn|✓|✓|✗|-|
> |SampleAttention|✓|✗|✗|-|
> |FastAttention|✗|✓|✗|-|
> |MOA|✗|✗|✗|-|
> |Reformer|✗|✓|✓|-|
> |**Ours**|**✓**|**✓**|**✓**|**✓**|
>
> ---
> >### Comment3
>
> **Reply**:
> We provide more video demos of CogvideoX and Open-Sora-Plan at: https://anonymous.4open.science/r/spa-videos/README.md.
>
> ---
> >### Question1
>
> **Reply**: Thank you for the reasonable suggestion. We compare the score of attention recall and the relative L2 distance of the final video/image outputs on CogvideoX and Stable-Diffusion-3.5. The results are shown in the following Tables.
>
> **Comparison on CogvideX:**
> |Attenion|Attention Recall|Relative L2 of outputs|
> |-|-|-|
> |Minference|0.862|0.228|
> |FlexPrefill|0.811|0.378|
> |SpargeAttn|0.892|0.056|
>
> **Comparison on Stable-Diffusion-3.5:**
> |Attenion|Attention Recall|Relative L2 of outputs|
> |-|-|-|
> |Minference|0.922|0.419|
> |FlexPrefill|0.854|0.464|
> |SpargeAttn|0.936|0.126|
>
> ---
> >### Question2
>
> **Reply**:
> Thank you for your insightful question. Simply splitting heads to different GPUs will result in a load balance problem. To address the problem, we can obtain the sparsity of each head in the tuning process and evenly distribute these heads according to the sparsities. We conduct a small experiment on a set of CogVideoX tensors:
>
> ||Simple Split to Two GPUs|Split According to Tuning Sparsity|
> |-|-|-|
> |GPU1-Latency|15.9ms|16.8ms|
> |GPU2-Latency|17.6ms|16.7ms|
> |Final Latency|17.6ms|16.8ms|
>
> Fortunately, the split method is compatible with Ulysses parallel method.
>
> ---
> >### Question3
>
> **Reply**:
> Thank you for the insightful question. We analyze the sparsity and sparse patterns across heads at **[Analysis Repo]**. There are heads with no sparsity, i.e., no acceleration, and heads with very high sparsity, i.e., high acceleration. However, it is not a serious problem generally. For example, for a set of Q, K, V in the shape of (2, 30, 16384, 64), it needs to launch 2 * 30 * 16384/128=7680 GPU blocks to do FlashAttention individually. A GPU usually only has about 100 SMs. The latency variance of blocks is not a serious problem because the overall throughput is determined by the average sparsity. A significant load imbalance may only occur in a multi-GPU environment, and this issue has already been addressed in the previous response.
>
> ---
> >### Question4
>
> **Reply**:
> We conduct an experiment on Open-Sora-Plan, and the result is as follows:
>
> |Attention|Sparsity↑|CLIPSIM↑|CLIP-T↑|VQA-a↑|VQA-t↑|FScore↑|End-to-end Latency↓|
> |-|-|-|-|-|-|-|-|
> |Original Attention|0|0.16503646|0.999496|81.40257|80.601264|0.84729|629s|
> |SpargeAttn|0.341|0.168645|0.99859|77.5948|76.9102|0.83938|393s|
>
> Also, we provide video demos of Open-Sora-Plan at https://anonymous.4open.science/r/spa-videos/README.md.
>
> ---
> If you feel your concerns have been resolved, we would greatly appreciate it if you consider raising the score.

---

> > ### Comment · Reviewer_m9qG · 2025-04-09
> >
> > The paper is earlier than STA(FastVideo). The rebuttal has provided substantial information that well addresses my primary concerns. The additional experiments may enhance the community's understanding of sparsity patterns in the current model. I am raising my score to ​3 (Weak Accept) with an inclination toward acceptance.

---

### Official Review · Reviewer_ciBv · 2025-03-15

**Overall Recommendation:** 3

**Summary:**

The paper proposes SpargeAttn, a universal and training-free sparse attention mechanism intended to accelerate inference across diverse models, including language, image, and video generation. SpargeAttn operates in two stages: initially, it rapidly predicts sparse regions of the attention map using selective token compression; subsequently, it employs a warp-level sparse softmax to further omit negligible computations without extra overhead. Experiments conducted on various benchmarks (including Llama3.1, CogvideoX, Mochi, Flux, and Stable-Diffusion3.5) suggest that SpargeAttn achieves significant speedups (up to 5x faster) without negatively impacting model accuracy or end-to-end performance.

**Claims And Evidence:**

Yes

**Essential References Not Discussed:**

[3] also leveraged the Z-order curve (another type of space-filling curve similar to the Hilbert curve) in sparse attention. It would be beneficial if the authors could include a discussion on this closely related work.

[3] ZETA: Leveraging Z-order Curves for Efficient Top-k Attention, ICLR 2025

**Experimental Designs Or Analyses:**

Yes.

**Methods And Evaluation Criteria:**

Yes

**Other Comments Or Suggestions:**

Typo: Line 237, ture -> true

**Other Strengths And Weaknesses:**

Strengths:

1. General applicability across various model types (language, image, video).

2. Effective speedups demonstrated empirically on a wide range of benchmarks.

3. Minimal overhead introduced in sparse attention prediction, especially beneficial for long sequences.

4. Practical implementation details, including integration with quantization methods, enhance its usability.

Weaknesses:

1. How robust is SpargeAttn when deployed on models with highly diverse or previously unseen attention patterns? NSA [1] employs three branches—Compression, Selection, and Sliding—to compensate for block-wise attention. In contrast, SparseAttn applies attention exclusively to selected blocks. Could this selective strategy lead to information loss?


2. What are the practical guidelines or strategies recommended for systematically selecting hyperparameters $(\tau, \theta, \lambda)$ in real-world deployment? The reliance on heuristic hyperparameter tuning $(\tau, \theta, \lambda)$ for optimal performance could limit straightforward generalization. Large attention models, in particular, require substantial training resources. The authors perform a grid search over hyperparameters for each attention layer, significantly increasing computational costs. Could the authors clarify how they address or mitigate these computational burdens?

3. The contribution appears primarily engineering-focused, with relatively incremental methodological advances. Could the authors clarify or elaborate on how their contributions distinctly advance beyond previous block-wise attention methods?

[1] Native Sparse Attention: Hardware-Aligned and Natively Trainable Sparse Attention, arXiv

**Questions For Authors:**

See weakness

**Relation To Broader Scientific Literature:**

This work is closely related to NSA [1] and MoBA [2]; however, both of these studies were published after ICML submission.

[1] Native Sparse Attention: Hardware-Aligned and Natively Trainable Sparse Attention, arXiv
[2] MoBA: Mixture of Block Attention for Long-Context LLMs, arXiv

**Theoretical Claims:**

No theoretical results are included.

---

> ### Author Rebuttal · Authors · 2025-03-31
>
> Dear Reviewer ciBv,
> Thank you for your valuable suggestions and questions. Below, we address each point raised.
>
> ---
> > **Essential References Not Discussed**
>
> **Reply.** Thank you so much for providing the reference. We will discuss it in our paper.
>
> ---
> > **W1.** How robust is SpargeAttn when deployed on models with highly diverse or previously unseen attention patterns? NSA [1] employs three branches—Compression, Selection, and Sliding—to compensate for block-wise attention. In contrast, SparseAttn applies attention exclusively to selected blocks. Could this selective strategy lead to information loss?
>
> **Reply**: Thank you for your question. We believe there may be a few misunderstandings:
>
> 1. **Attention Patterns**: *SparseAttn* is a test-time sparse attention method that does not rely on previously unseen patterns.
> 2. **Training vs. Inference**: NSA requires model retraining, while *SparseAttn* is applied directly during inference.
> 3. **Robustness**: All sparse attention for inference will loss some information.  However, *SparseAttn* is robust because it is error-bounded and dynamically adjusts sparsity per input to maintain accuracy. For example, if an input lacks sparsity, *SparseAttn* predicts zero sparsity and skips no computations.
>
> Finally, we directly compare *SparseAttn* and NSA for inference on Llama3.1-8B, with results shown in the table:
>
> |Attention|Speed (TOPS) ↑|WikiText (Ppl.) ↓|Longbench ↑|NIAH ↑|
> |-|-|-|-|-|
> |Full-Attention|156.9|6.013|38.682|0.907|
> |NSA|1098.8|78.335|7.496|0.07|
> |SpargeAttn|708.1|6.02|39.058|0.909|
>
>
> ---
> > **W2.** What are the practical guidelines or strategies recommended for systematically selecting hyperparameters ($\tau, \theta, \lambda$) in real-world deployment? The reliance on heuristic hyperparameter tuning ($\tau, \theta, \lambda$) for optimal performance could limit straightforward generalization. Large attention models, in particular, require substantial training resources. The authors perform a grid search over hyperparameters for each attention layer, significantly increasing computational costs. Could the authors clarify how they address or mitigate these computational burdens?
>
> **Reply**: Thank you for the vital advice.
>
> First, we can use fixed hyperparameters (τ=0.5, θ=0.95, λ=-25) for inference without tuning. Although the sparsity is not as high as tuning, it is a convenient way. The results on CogVideo are presented in the Table:
>
> |Attention|Speed (TOPS) ↑|CLIPSIM ↑|CLIP-T ↑|VQA-a ↑|VQA-t ↑|FScore ↑|
> |-|-|-|-|-|-|-|
> |Full-Attention|166|0.1819|0.9976|80.384|75.946|5.342|
> |SpargeAttn|402.39|0.1802|0.9974|79.416|74.931|5.104|
>
> Second, we would like to clarify that the tuning phase doesn't require training - it only needs 5-10 inference passes. On a server with 8×4090 GPUs:
>
>  - Llama-3.1-8B takes just 14 minutes to tune.
>  - CogVideoX requires 2.6 hours.
>    Once tuned, the model can be used permanently for inference. We will also release pre-tuned models for immediate use.
>
>
> ---
> > **W3.** The contribution appears primarily engineering-focused, with relatively incremental methodological advances. Could the authors clarify or elaborate on how their contributions distinctly advance beyond previous block-wise attention methods?
>
> **Reply**: Thank you for your suggestion, and we summarize our contributions:
> **Key Contributions:**
>
> 1. **Effectiveness**: We design the first sparse attention method to **accelerate across language, image, and video models** without compromising accuracy.
>
> 2. **Method Innovations**:
>  - First to enable block-wise sparse computation via selective compression.
>  - First to propose sparse online softmax (a fundamentally novel approach).
>  - First to establish guaranteed error bounds for all attention layers in the model.
>
> Moreover, we summarize some representative methods from four aspects, including whether needing a training process, whether relying on specific attention map patterns, whether being applicable to all models (language and diffusion), and whether implementing attention quantization:
>
> |Method|Training Free|Pattern Free|Universal|Quantization|
> |-|-|-|-|-|
> |MInference|✓|✓|✓|-|
> |DuoAttention|✗|✗|✗|-|
> |SeerAttention|✗|✓|✗|-|
> |FlexPrefill|✓|✓|✓|-|
> |InfLLM|✓|✗|✗|-|
> |SparQAttn|✓|✓|✗|-|
> |LokiAttn|✓|✓|✗|-|
> |MOA|✗|✗|✗|-|
> |**Ours**|**✓**|**✓**|**✓**|**✓**|
>
> ---
> If you feel your concerns have been resolved, we would greatly appreciate it if you consider raising the score.

---

### Official Review · Reviewer_P2oG · 2025-03-18

**Overall Recommendation:** 4

**Summary:**

This paper proposes a universal sparse attention mechanism that ensures both speedup and end-to-end performance of diverse models. Specifically, the method adopts a two-stage filtering schemed: In the first stage, it computes attention based on compressed tokens of self-similar blocks of query and key, and skip the computation of entries that have low cumulative sums; In the second stage, the method further identify small enough values in the attention map, and skip the negligible values. The experimental results show that the proposed method is able to maintain end-to-end metrics performance to full attention, while achieving significantly faster processing speed in diverse set of tasks.

**Claims And Evidence:**

The sparsity prediction mechanism proposed in the paper makes intuitive sense, and its strength is supported by the strong performance / processing speed trade off over related sparse attention baselines in experiment section.

**Essential References Not Discussed:**

I haven't found missing essential references

**Experimental Designs Or Analyses:**

Yes, I checked all the subsections of the experiment section, the design of the experiments are valid and the results positively support the ideas of the paper.

**Methods And Evaluation Criteria:**

The benchmark datasets cover a wide range of tasks, supporting the universality claim of the proposed method, and for each task, the metrics used are suitable to measure the performance. The paper uses speed and sparsity to measure the computational efficiency of the proposed method, which also makes sense.

**Other Comments Or Suggestions:**

please see questions

**Other Strengths And Weaknesses:**

1. The paper is well written and easy to follow

**Questions For Authors:**

1. Since the attention computation among tokens is permutation invariant, why would using hilbertcurve permutation method result in a slightly worse precision comparing to other permutation variants as shown in Table 9 in the supplementary material?

**Relation To Broader Scientific Literature:**

The token tiling strategy used in this paper is adopted from flashattention [1],

[1] Dao, Tri. "Flashattention-2: Faster attention with better parallelism and work partitioning." arXiv preprint arXiv:2307.08691 (2023).

**Theoretical Claims:**

There are no theoretical claims in this paper

---

> ### Author Rebuttal · Authors · 2025-03-31
>
> Dear Reviewer P2oG,
> Thank you for your valuable suggestions and questions. Below, we address each point raised.
>
> ---
> > **Q1.** Since the attention computation among tokens is permutation invariant, why would using the Hilbertcurve permutation method result in a slightly worse precision compared to other permutation variants, as shown in Table 9 in the supplementary material?
>
> **Reply**:
> Thank you for your question. Yes, standard attention is token permutation invariant. However, we use sparse attention instead. After HilbertCurve permutation, the similarity in blocks of Q, K, and V increases, which enhances attention sparsity. This allows us to omit more computations, slightly increasing errors but improving speed.
>
> An Example:
>
> *Without permutation:*
>
> - Self-similar blocks: 80% (sparsity within self-similar blocks = 0.3)
> - Non-similar blocks: 20% (always computed in attention)
>   → **Effective sparsity**: 0.3 × 0.8 = 0.24
>   → **Blocks computed**: (1 - 0.3) × 0.8 + 0.2 = 0.76
>
> *With permutation:*
>
> - Self-similar blocks: 90% (sparsity within self-similar blocks = 0.3)
> - Non-similar blocks: 10% (always computed  in attention)
>   → **Effective sparsity**: 0.3 × 0.9 = 0.27
>   → **Blocks computed**: (1 - 0.3) × 0.9 + 0.1 = 0.73
>
> Additionally, we analyze and plot the sparsity and error brought by Hilbertcurve permutation, Random permutation, and Row-major permutation on CogvideoX tensors at this Link: https://anonymous.4open.science/r/tmp-442D/sparsity_error_of_permutation.pdf. It can be observed that the Hilbertcurve permutation achieves the highest sparsity under the same error.
>
> ---
> >Last but importantly, the Hilbertcurve permutation is a relatively minor aspect of our work. The **key contributions** of SpargeAttn are:
>
> 1. **Effectiveness**: We design the first sparse attention method that can actually accelerate **across language, image, and video models** without compromising accuracy.
>
> 2. **Method Innovations**:
> - First to enable block-wise sparse computation via selective compression.
> - First to propose sparse online softmax (a fundamentally novel approach).
> - First to establish guaranteed error bounds for all attention layers in the model.
>
> Moreover, we summarize some representative methods from four aspects, including whether needing a training process, whether relying on specific attention map patterns, whether being applicable to all models (language and diffusion), and whether implementing attention quantization:
>
> |Method|Training Free|Pattern Free|Universal|Quantization|
> |-|-|-|-|-|
> |MInference|✓|✓|✓|-|
> |DuoAttention|✗|✗|✗|-|
> |SeerAttention|✗|✓|✗|-|
> |FlexPrefill|✓|✓|✓|-|
> |H2O|✓|✗|✗|-|
> |InfLLM|✓|✗|✗|-|
> |DitFastAttn|✓|✗|✗|-|
> |SparQAttn|✓|✓|✗|-|
> |LokiAttn|✓|✓|✗|-|
> |SampleAttention|✓|✗|✗|-|
> |FastAttention|✗|✓|✗|-|
> |MOA|✗|✗|✗|-|
> |Reformer|✗|✓|✓|-|
> |**Ours**|**✓**|**✓**|**✓**|**✓**|
>
> ---
> ---
> If you feel your concerns have been resolved, we would greatly appreciate it if you consider raising the score.

---

### Decision · Program_Chairs · 2025-05-01

**Decision:**

Accept (poster)

**Comment:**

This is my summary of the strengths and weaknesses of this submission.

- Strengths:
  - Really elaborate empirical evaluation across various transformer-based models and modalities such as text, image, and video, highlighting both the speedup in the attention step (up to 5$\times$) as well as the ability to match the predictive performance of full attention. The authors also provided various additional evaluations and empirical analyses as per reviewer suggestions, and all the results provide useful intuition about the sparsity patterns, and highlight the robustness of the proposed scheme.
  - The sparsity pattern is adaptive, allowing the system to automatically pick the sparsity level as needed, and can work in a training-free manner, thereby making this an easy drop-in replacement in existing pretrained models for improved inference speed without loss in predictive performance.
- Weaknesses:
  - The hyperparameters introduced in this setup could require tuning to get optimal speed-accuracy tradeoff. However, the authors mentioned in their responses that (i) There is a default value that still provides a strong tradeoff, and (ii) The tuning of these hyperparameters only requires some number of inference passes, which take relatively less amount of time.
- The authors mention that error in the attention layers is guaranteed to be bounded. However, I do not see any theoretical analysis or derivation showing this bound. This needs to be clarified.

I would strongly suggest that the submission incorporates the clarifications of the reviewer confusion and misunderstanding in the revision of the paper.